# A Dual-Channel Microfluidic Chip for Single Tobacco Protoplast Isolation and Dynamic Capture

**DOI:** 10.3390/mi13122109

**Published:** 2022-11-29

**Authors:** Huali Zhang, Qianqian Geng, Zhanghua Sun, Xiaoxiang Zhong, Ying Yang, Shuangyu Zhang, Ying Li, Yali Zhang, Lijun Sun

**Affiliations:** 1School of Mechanical Engineering, Nantong University, Nantong 226019, China; 2School of Life Sciences, Nantong University, Nantong 226019, China; 3School of Medicine, Nantong University, Qixiu Road 19, Nantong 226001, China

**Keywords:** dual-channel microfluidics chip, isolation and dynamic capture, tobacco protoplasts

## Abstract

Protoplasts are widely used in gene function verification, subcellular localization, and single-cell sequencing because of their complete physiological activities. The traditional methods based on tissues and organs cannot satisfy the requirement. Therefore, the isolation and capture of a single protoplast are most important to these studies. In this study, a dual-channel microfluidic chip based on PDMS with multi-capture cavities was designed. The design theory of the dual-channel microfluidic chip’s geometry was discussed. The capture mechanism of the single cell in a dual-channel microfluidic chip was studied by simulation analysis. Our results showed that a single polystyrene microsphere or tobacco protoplast was successfully isolated and trapped in this chip. The capture efficiency of the chip was 83.33% for the single tobacco protoplast when the inlet flow rate was 0.75 μL/min. In addition, the dynamic capture of the polystyrene microsphere and tobacco protoplasts was also presented. Overall, our study not only provided a new strategy for the subsequent high throughput single protoplast research, but also laid a theoretical foundation for the capture mechanism of the single cell.

## 1. Introduction

Due to the complete physiological metabolic process, the protoplasts of plant cells are widely used in subcellular localization, protein interaction [1], gene expression and regulation [2], evaluation of gene-directed mutation rate [3], and synthesis of secondary metabolites. For example, combined with a plant hormone reporter gene, the changes of plant hormone signal in the protoplasts can be quickly explored with a high throughput [4]. Transient gene expression in the protoplasts can be achieved by the electroporation or induction of polyethylene glycol (PEG) [5,6]. In addition, the fusion of different protoplasts can cope with the defect of interspecific incompatibility, creating various homokaryon or heterokaryon types [7]. More important, due to the heterogeneity of intercellular transcriptome, single-cell transcriptome sequencing is becoming more popular, which can quantify transcriptional activity at the single-cell level [8]. Therefore, the isolation and capture of a single protoplast are becoming more important.

For the isolation and capture of a single cell, microfluidic devices are widely used as powerful tools, especially for various types of animal or human cells [9,10,11]. There are some methods for single-cell trapping in microfluidics such as electric filed force [12], dielectric power [13], magnetic force [14], acoustic force [15], optical tweezer [16], and hydrodynamic trapping [17]. Compared with other manipulation methods, hydrodynamic trapping provides great flexibility with respect to the structure design, easy operation, and high cell integrity. For instance, Liu et al. (2018) developed an S-shaped loop channel with trap units [18]. A hydrodynamic force was used in a micro-cup structure side-by-side in the channel to achieve high-throughput single-particle trapping with a flow ratio of 1.744. These studies failed to use live cells for the trapping experiments. Ying Li et al. (2020) developed an injection molding method to fabricate microcup-shaped microfluidic chips for single-cell capture with a capture efficiency of about 80% [19]. Dongguo Lin et al. (2020) designed a microscreen array microfluidic chip for screening breast cancer-specific therapeutic agents, which had an excessively low efficiency of capturing cancer cells at 60% [20]. The physical and chemical properties of cells and particles are very different, and the cells are prone to deformation and escaping from the capture chamber during the capture process, resulting in a decrease in the capture efficiency of the microfluidic chip [21]. The geometric effects on the flow behavior of Hela cells in a double-slit microfluidic chip were investigated to enhance the trapping of Hela cells, while the efficiency of cell capture is relatively low from the experimental results [22]. A dual-well high-yield cell trapping device was developed to study the spreading, proliferation, and differentiation of KT98 mouse neural stem cells [23]. This microporous structure of the chip relied on the gravity of the cell to achieve the capture, which lead to a long capture time, and it was easy to miss the detection of important substances in the cell. In addition, a passive microfluidic device enabling both cell trapping and releasing in a deterministic order was reported, which contained an array of trapping sites, connected to a capillary burst valve for trapping and a capillary trigger valve for MCF-7cell releasing [24].

Although significant progress has been made in single animal or human cell capture based on the hydrodynamic trapping structures, these micro-chip devices for single cell capture have relatively complicated geometries, the high cost of fabrication, and low trapping efficiency, which cannot meet the live cellular analysis. In addition, there was no application for microfluidic trapping of single protoplast of plants. In this study, a simple dual-channel microfluidic device based on hydromechanics was constructed to observe the dynamic trapping process of the single polystyrene microsphere or tobacco protoplast. The advantages of single-channel chips are the simple structure, they are easy to manufacture, and they have a low manufacturing cost. It does not require a variety of control and observation instruments, just fluidic power to carry the cells. Compared with other structures of the chip, it avoids the damage of various operating instruments to the cell such as electricity, magnetism, laser, sound and so on.

## 2. Experimental

### 2.1. Reagents and Materials

PDMS silgard 184 was purchased from Dow Corning (Midland, MI, USA). SU-8 2035 photoresist was purchased from the Gersteltec (Vaud, Switzerland). Silicon chips with a thickness of 360 μm were bought from Suzhou double metal materials Co., Ltd. The polystyrene microbeads with a diameter of 30 μm were purchased from Yuan Biotech (Shanghai, China). Additionally, 1 mL syringes were bought from Nantong’s biological experiment equipment Co., Ltd. (Nantong, China). Other reagents were of analytical grade.

### 2.2. Microchip Fabrication

The glass-encapsulated PDMS chips were fabricated using a standard soft lithography technique, as reported previously [25]. Briefly, the capture microfluidic channel was designed by drawing software (CAD) and printed on transparency film (Figure 1A). Then, transparency film was used as a photomask for coating the silicon wafer covered with SU-8 2035 photoresist, which was exposed to EXFO OmniCurc^®^ S1000 ultraviolet point light to form the capture microfluidic channel. PDMS Sylgard 184 (mixing in the ratio of 10:1) was poured onto the mold (the silicon wafer surface with the consolidated channels) and heated. After baking, punch two holes on the back of the glue with corresponding punchers with a diameter of 1 mm, which were used for the injection of the solution of protoplasts (1) and outlet for solution (2). PDMS and a glass slide were required to be cleaned by using the PDC-32GB oxygen plasma and then firmly pasted them together (Figure 1B). Figure 1C showed the model of the capture microfluidic channel. Two channels with a total 12 capture chambers were constructed. Images of the microbeads or tobacco protoplasts were taken on a LeicaDMi 3000b fluorescent inverted microscope (Wetzlar, Germany). Teflon tubing was connected to the inlets of the channels for transporting microbeads or tobacco protoplasts by langer LSP02-1B double-channel injection pump (Nanjing, China) (Figure 1D).

### 2.3. Extraction Process of Tobacco Protoplasts

For obtaining the protoplasts, the sun’s method was consulted [26]. Briefly, the seeds of tobacco were planted in 1/2 MS culture medium in a climate-controlled incubator with a 14 h light/10 h dark photoperiod at 26 °C/24 °C for 6 weeks. The sterile leaves of tobacco were selected and cut into samples with a width of 1 mm. Then, samples of leaves were put into the sterilized NP solution (15 mg/mL cellulose, 4 mg/mL pectase, 4 mM morpholineethanesulfonic acid, 0.8 M CaCl_2_·2H_2_O, 2 M KCl, and 0.5 M mannitol, pH 5.8) for enzymolysis at 25 °C about 5 h. After that, the solution containing protoplasts was required to centrifuge for 6 min at 150 rcf. Then the protoplast precipitation was resuspended by the modified W5 solution (0.16 M NaCl, 0.09 M CaCl_2_·2H_2_O, 0.5 mM KCl and 4 mM morpholineethanesulfonic acid, pH 5.8), and washed three times by centrifugation to discard the enzyme. Finally, the protoplasts were resuspended by NP that contained 0.25 mg/L kinetin and 2 mg/L 2,4-dichlorophenoxyacetic acid (2,4-D).

## 3. Results and Discussion

### 3.1. Theoretical Analysis of Channel Design

According to the previous reports [26,27,28], three assumptions in the channel design were also considered. First, the fluid is pressure-driven and laminar. Second, the fluid is regarded as a non-compressive and Newtonian fluid. Third, the flow velocity distribution is parabolic. Therefore, the theoretical design of the channel was discussed as shown in Figure 2A,B.

S and S_a_ are the cross-sectional areas of the corresponding channels and h is the height of the channel. *W_a_** h* *V_a_* = S_a_* *V_a_* = *Q_a_*, *W** h* *V* = S* *V* = *Q*, *W_a_*/*W* = (*V** *W_a_** h* *V_a_*)/(*V_a_** *W** h* *V*). Therefore, the ratio of the virtual width (*W_a_*) to the main channel width (*W*) is given [27]:(1)$$\frac{W_{a}}{W}=\frac{V\times Q_{a}}{V_{a}\times Q}$$
where *Q* is the volume rate of flow and *V* was the average velocity of the flow in the stream. The subscript a or no subscript refers to the virtual stream or the total stream, respectively. The velocity profile inside the main channel *u* (*y*) can be expressed as follows [28,29]:(2)$$u\left(y\right)=\frac{1}{2\mu }\left(-\frac{dp}{dx}\right)(Wy-y^{2})$$
where *μ* is the viscosity of the fluid, *dp*/*dx* refers to the pressure gradient at both ends of the channels, and 0 ≤
*y*
≤
*W*. The ratio *W*_a_/*W* can be estimated from the above two equations, as shown below [27]:(3)$$2{\left(\frac{W_{a}}{W}\right)}^{3}-3{\left(\frac{W_{a}}{W}\right)}^{2}+\frac{Q_{a}}{Q}=0$$

In the case of the virtual width after a single particle was trapped (*W_a_*), and the virtual width before trapping (*W_b_*), the virtual width should be satisfied with the following in-equations [28]:(4)$$W_{b}<r,W_{a}>r$$

According to the law of Hagen and Poiseu, the circuit of fluid can be explained by the theory of the circuit. A further formula derivation process can be seen in [28]. In short, for a rectangular bypass channel, the flow resistance *R_s_* is given by
(5)RS=fRe×μ×L2Dh2×A
where *f*_Re_ is the friction coefficient of laminar flow, which depends on the width-to-depth ratio of the channel, *L* is the length of the bypass, that is, (2*L_w_* + *L_h_*)_,_
*D_h_* is the hydraulic diameter of the channel and *A* is the cross-section of the channel. For irregular trapping cavity, the flow resistance *R_a_* can be calculated as follow [28]:(6)$$R_{a}=2\mu \times L_{h}\times \frac{P_{a}^{2}}{A_{a}^{3}}$$
where *L_h_* is the trap length, *P_a_* and *A_a_* are the perimeter of the trap and cross-sectional area of the trap, respectively. A design method for single-cell trapping is then obtained by substituting Equations (5) and (6) into in-Equation (4) [28]. According to the diameter of the polystyrene microbeads or protoplasts (about 20~30 μm) in our study, the design dimensions of the trapping channels were determined and listed in Table 1.

### 3.2. Simulation Analysis of the Single-Cell Trapping

For simulating the fluid flows during single-cell trapping, FLUENT module in ANSYS Workbench software was used to build the 3D model of our capture channels. The simulation software we used was ANSYS 2020 R2 (The number of meshes is 1,002,764, and the minimum size is 3 μm, which meets the requirements of simulation calculation). The inlet boundary condition in the simulation was set to the velocity of 0.0026 m/s at the entrance of Channel in Figure 3A,B (whereas the flow rate of the pump was 0.75 µL/min, and the cross-sectional area of the entrance was calculated as 0.0048 mm^2^). The outlet boundary condition was set as pressure (atmospheric pressure). Pressure changes in the channel with an empty cavity and in the channel with a trapped microbead were illustrated in Figure 3A,B. Seven locations were selected from each corresponding position, marked as Ⅰ, Ⅱ, Ⅲ, Ⅳ, Ⅴ, Ⅵ, and Ⅶ. The pressures of five sites for each location were collected to obtain the change in the pressure. It could be seen that the pressures at the same location in the channel were significantly different (Figure 3C). The pressures of Ⅰ–Ⅳ in the channel in Figure 3B were significantly higher than that of channel A when a single microbead was trapped, which indicated that other microbeads could not enter the capture cavity further. In addition, the pressures of Ⅱ–Ⅳ around the entrance of the trapping cavities were greater than that of Ⅴ–Ⅵ. An extrusion pressure will be formed to squeeze the microbeads into the bypass channels, and eventually fix the microbeads in the trapping cavities. Moreover, as the pressures in the bypass channel significantly decreased from Ⅴ, Ⅵ, to Ⅶ, back pressure can be provided in the trapping cavities to trap the microbeads inside. In addition, the pressure variations in these seven points were not enough to capture excess cells except for one single cell. The pressures in point Ⅶ for Channel in Figure 3A,B were equal, which indicated that the entrance pressure for the next channel would be stable. The chip has long main channels of the chip provided enough entrance pressures for all trapping channels. Therefore, our simulation is very important to fabricate microfluidic trapping chip because it verified our conjecture of cell capture by multi-channels chip we designed.

### 3.3. The Capture Performance of the Dual-Channel Microfluidic Chip

The polystyrene microbeads with a diameter of 20–25 μm were used to test the capture performance of the dual-channel microfluidic chip. The results showed that the cavity structure chip provided a reliable capture for the microbeads (Figure 4). Compared with the reported microfluidic chip for trapping microbeads [27,30], our cavity structures provided enough space for further cell culture and single-cell analysis in vitro. Although there are many pipeline capture chips, most of them have complex structures and are difficult to manufacture, which require high environment and consumables for chip manufacturing. The dynamic capture of the microbeads was also shown in the Appendix A. The video presented the microbeads entered the microchip from the left entrance and were captured successively.

The microfluidic chip consists of two arrays of chambers, and each chamber contains six channels with their corresponding capture cavities (Figure 5A). To study the effect of the flow rate on the manipulation of the protoplasts, tests at gradient flow rates of the inlet were conducted with the syringe pump. The results showed that the low flow rate (<0.5 μL/min) may cause the blockage of the flow channel and aggregation of the protoplasts in the cavities. Protoplasts may be blocked in the channel at a low flow rate because the protoplasts moved slowly in the channels due to insufficient hydraulic thrust. Therefore, some protoplasts in the channel may contact each other in this low flow, resulting in the blockage of the channel and the capture chamber. However, the cell membranes of the protoplasts were squeezed by the cavity walls at the flow rate of 1 μL/min, which lead to the rupture or deformation to the single protoplasts. The excessive velocities could affect the mechanical properties of live cells in microfluidic systems, as demonstrated previously [31]. The flow rate at 0.75 μL/min resulted in a significant improvement in the capture efficiency and the integrity of the tobacco protoplasts in our device, which is sufficiently stable for single-cell trapping applications. As shown in Figure 5B, after injecting the tobacco protoplasts suspensions, ten single protoplasts were found trapped in 10 single cavities and the other two channels were blocked. The corresponding cell capture efficiency was 83.33% (which is the ratio of single cells trapped to the total number of capture cavities). As shown in Figure 6, an integrated protoplast was captured in one single channel. The dynamic capture of the tobacco protoplasts was also shown in the Appendix A. The video presented the tobacco protoplasts entered the microchip from the left entrance at the flow rate at 0.75 μL/min and were captured successively. These results showed that our designed chip could be used for single-cell trapping which could provide effective methods to study the cell culture, division, and cell wall regeneration for a single cell.

## 4. Conclusions

The isolation and capture of a single protoplast are most important to gene function verification, subcellular localization, and single-cell sequencing. In this study, a dual-channel microfluidic chip based on PDMS with multi-capture cavities was designed. The theory was discussed to design the geometry of this chip. The capture mechanism of the single cell in dual-channel microfluidic chip was studied by simulation analysis. Our results showed that a single polystyrene microsphere or tobacco protoplast was successfully isolated and trapped in this chip. The precise design of the capture cavities and control of the inlet velocity maintained the trapped cells more stable in the cavities. The capture efficiency of the chip was 83.33% for the single tobacco protoplast when the inlet flow rate was 0.75 μL/min. In addition, the dynamic capture of the polystyrene microsphere and tobacco protoplasts was also presented. Therefore, this study not only provides a new strategy for the subsequent high throughput of single protoplast research, but also lay a theoretical foundation for the capture mechanism of the single cell. The single-channel chip successfully captured protoplasmic cells in 10 of the 12 capture cavities. The single-channel structure we designed can improve the capture capability of the chip by adding more channels. Different types of animal and plant cells of different sizes can be captured by designing different sized capturing cavities.

## Figures and Tables

**Figure 1 micromachines-13-02109-f001:**
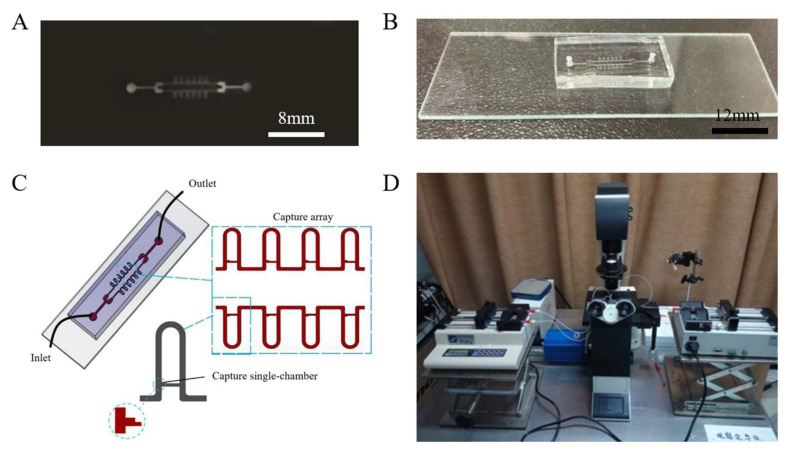
Fabrication and characterization of the capture microfluidic chip. (**A**) Silicon wafer (**B**) Image of a fabricated chip (**C**) Schematic illustration of the capture chip with 2 arrays of 12 chambers (**D**) Observation through the fluorescence microscope.

**Figure 2 micromachines-13-02109-f002:**
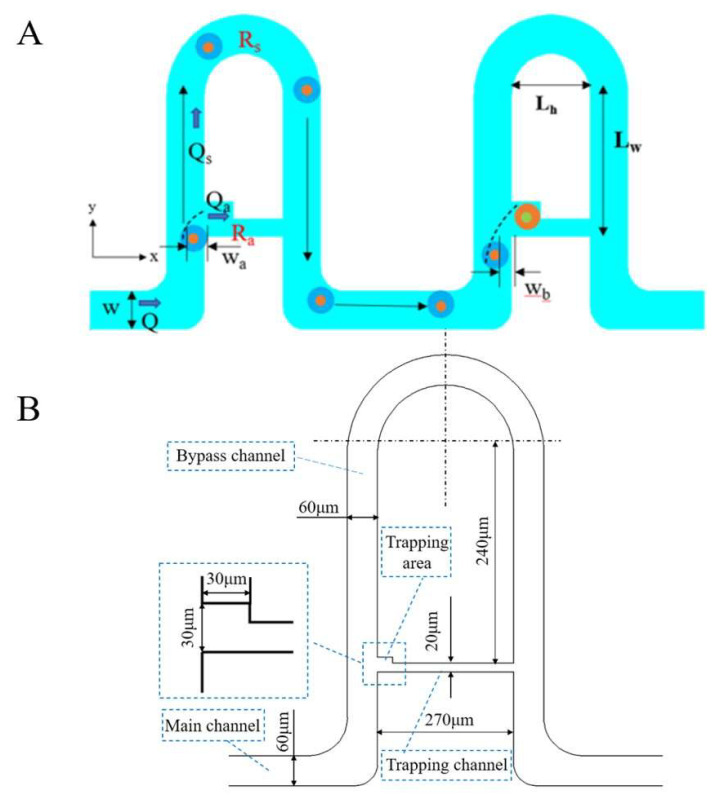
(**A**,**B**) Mathematical model of the trapping channels.

**Figure 3 micromachines-13-02109-f003:**
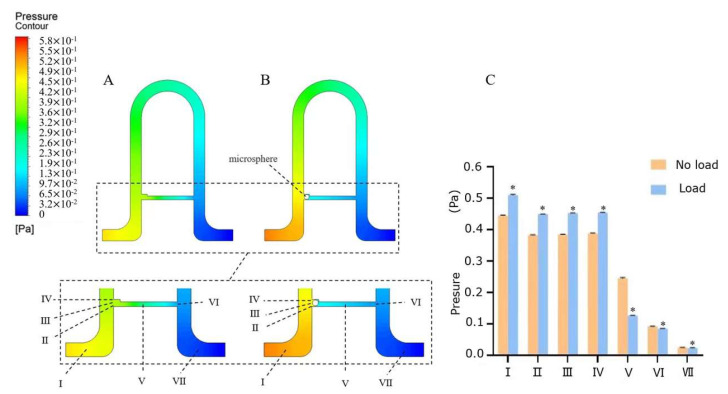
Simulated pressure characteristics of the trapping cavities and main channel. (**A**) Pressure contour of empty capture area and bypass channel at the flow velocity of 0.0026 m/s. (**B**) Pressure contour of Channel with a trapped microbead. (**C**) Calculated pressure variation for channel A and Channel B, * indicates significant difference *p* values ≤ 0.05.

**Figure 4 micromachines-13-02109-f004:**
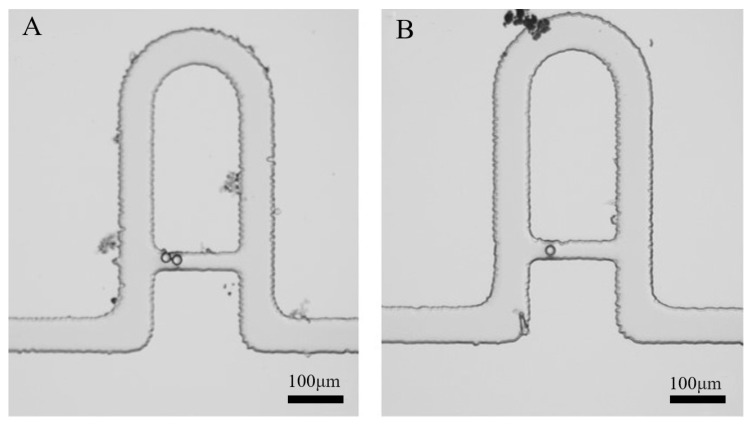
Micrographs of 20–25μm Polystyrene Microbeads trapped in cavities. (**A**) One microbead trapped in a capturing cavity (**B**) two microbeads tapped in a capturing cavity.

**Figure 5 micromachines-13-02109-f005:**
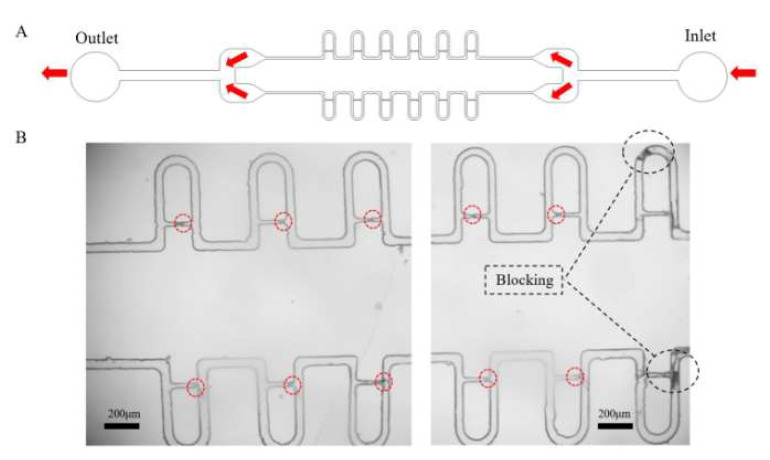
(**A**) The illustrations of the capture chip with 12 trapping cavities. (**B**) Microscopic image of single live tobacco protoplasts trapped in the cavities.

**Figure 6 micromachines-13-02109-f006:**
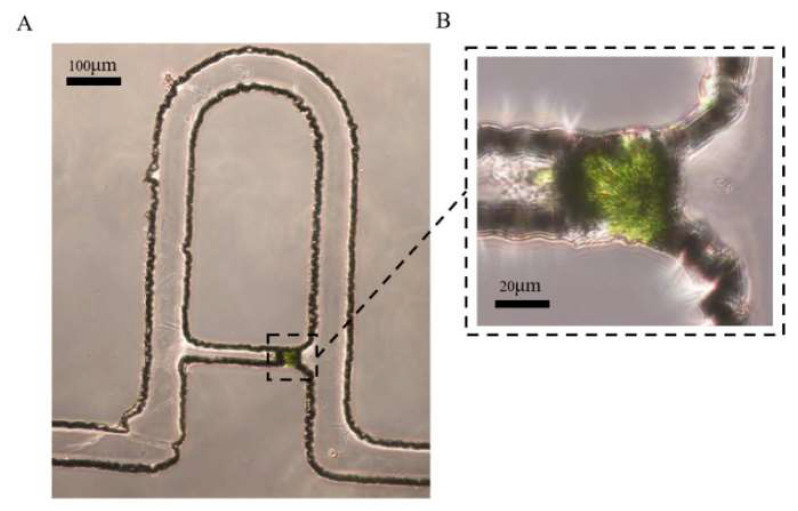
(**A**) Enlarged microscopic image of one single tobacco protoplast trapped. (**B**) Partial enlargement of figure A.

**Table 1 micromachines-13-02109-t001:** The dimensions of the chip.

Main Channel Width, *W*	Trapping Area, *A_a_*	Trap Length, *L_h_*	Bypass Length, *L_w_*	Depth of Main Channel	By Pass Channel
60 μm	30 × 30 μm^2^	270 μm	440 μm	80~100 μm	20 μm

## Data Availability

The datasets generated during and analyzing the current study are available from the corresponding author on reasonable request.

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
