# Peer review of "A Dual-Channel Microfluidic Chip for Single Tobacco Protoplast Isolation and Dynamic Capture"

_micromachines, 2022, doi:10.3390/mi13122109_

Round 1

Reviewer 1 Report

In this paper, authors have studied the isolation and capture of single protoplast to gene function verification, subcellular localization, and single-cell sequencing in a dual-channel microfluidic chip based on PDMS with capable of multi-capture cavities. The design and geometry of this chip was theoretically investigated, and simulation analysis was applied to examine the single cell trapping mechanism. This article can be reconsidered for publication after addressing the following issues:

1.        Introduction section: Some more examples of the recent studies need to be explained and cited to compare the recent study with them.

2.        Equations used throughout the manuscript are not cited. Please cite them.

3.        ANSYS software is used for the simulation. But which version and number are used is missing and needs to be mentioned.

4.        Meshing done for the simulation is not considered whereas it plays an important role. Mesh analysis must be used and then the simulation results can be considered for further analysis.

5.        In “theoretical analysis of channel design”, it can be seen that the authors have followed references 24, 25, and 26 for the basic theoretical derivations. It is suggested that theoretical derivations need to be discussed with the basic equations to derive the ratio of virtual width in more detail; it would be more understandable to the readers.

6.        The authors should carefully check the terms and nomenclatures, for example in page 4, “Where u is the viscosity of the fluid”, here “µ is the viscosity of the fluid” suggested checking the entire document.

7.        The idea of this article is innovative and appreciated. Need some focus on Figure 1, 3, 4, and 5, which is not co-relatable. The result with the live tobacco protoplasts tapper chip seems not same as described in figure 1 and the simulation-designed chip. Suggested to respective image with live cell, as Figure 1 or explanations needed how possible these results can be compared.

8. On page 7 “The results showed that the lower flow rate (<0.5 μl/min) may cause the blockage of the flow channel and aggregation of the protoplasts in the cavities” authors should differentiate this point to which results are they discussed about. 

Author Response

Response to reviewer

  1. Introduction section: Some more examples of the recent studies need to be explained and cited to compare the recent study with them.

Response 1: Thank you for your thoughtful suggestion. We have added the recent studies (Reference number 19,20) in the introduction section (to see the line 51 to 59 in page 2, and marked in blue).

  1. Equations used throughout the manuscript are not cited. Please cite them.

Response 2: Thank you for your thoughtful suggestion. We have cited each equation (to see Equations 1 to 6 on the 4 and 5 pages, and marked in blue) in the manuscript.

  1. ANSYS software is used for the simulation. But which version and number are used is missing and needs to be mentioned.

Response 3: Thank you for your thoughtful suggestion. The simulation software we uses is ANSYS 2020 R2 and has been described in the manuscript (to see the line 168 to 170 in page 5 and marked in blue).

  1. Meshing done for the simulation is not considered whereas it plays an important role. Mesh analysis must be used and then the simulation results can be considered for further analysis.

Response 4: Thank you for your thoughtful suggestion. The number of meshes is 1002764, the minimum size is 3μm ,which meets the requirements of simulation calculation(to see the line 169 to 170 in page 5 and marked in blue) and the simulation file package is included in the attachment for the reader to check.

  1. In “theoretical analysis of channel design”, it can be seen that the authors have followed references 24, 25, and 26 for the basic theoretical derivations. It is suggested that theoretical derivations need to be discussed with the basic equations to derive the ratio of virtual width in more detail; it would be more understandable to the readers.

Response 5: Thank you for your thoughtful suggestion. The ratio of virtual widths has been derived in detail for your proposal as follows (to see the line 133 to 136 in page 4 and marked in blue )

S and Sa are the cross-sectional areas of the corresponding channels, and h is the height of the channel. Wa*h*Va=Sa* Va= Qa, W*h*V=S*V=Q, Wa/W=(V*Wa*h*Va)/ (Va*W*h*V). Therefore, the ratio of the virtual width (Wa) to the main channel width (W) is given [27].

  1. The authors should carefully check the terms and nomenclatures, for example in page 4, “Where u is the viscosity of the fluid”, here “µ is the viscosity of the fluid” suggested checking the entire document.

Response 6: Thank you for your thoughtful suggestion. Here u is wrong, µ is correct, we have checked the manuscript and the error has been corrected (to see the line 143 in page 4 and marked in red)

  1. The idea of this article is innovative and appreciated. Need some focus on Figure 1, 3, 4, and 5, which is not co-relatable. The result with the live tobacco protoplasts tapper chip seems not same as described in figure 1 and the simulation-designed chip. Suggested to respective image with live cell, as Figure 1 or explanations needed how possible these results can be compared.

Response 7: Thank you for your thoughtful suggestion. Figure 1 (Figure 2 in the current manuscript) is a theoretical model of the capture channel, expressing the geometry of the capture channel and the flow of cells in the channel. Figure 3 (Figure 4 in the current manuscript) shows the particle capture experiments done in the theoretical model channel of Figure 1. Figure 4 (Figure 5 in the current manuscript) is a chip channel with the same structure as Figure 1 and Figure 2, and the capture of tobacco protoplasts was done in the channel. Figure 5 (Figure 6 in the current manuscript) is an enlarged view of a localized individual capture channel of Figure 4. These figures are related.

  1. Onpage 7 “The results showed that the lower flow rate (<0.5 μl/min) may cause the blockage of the flow channel and aggregation of the protoplasts in the cavities” authors should differentiate this point to which results are they discussed about. 

Response 8: Thank you for your thoughtful suggestion. The corresponding discussion of the lower flow rate is added in the manuscript (to the line 218 to 221 in page 7 and marked in blue).

Protoplasts may be blocked in the channel at low flow rate because the insufficient hydraulic thrust on the protoplasts causes the protoplasts to move slowly in the channel. Therefore, some protoplasts in the channel may contact with each other in this low fluid flow, resulting in the blockage of the channel and the capture chamber.

Reviewer 2 Report

I read this manuscript with interest. The following comments may help the authors to improve the manuscript before acceptance.

1.       Schemes 1C and 1D are so small. They need to be enlarged.

2.       Reference for equation 2 (e.g. https://link.springer.com/article/10.1007/s10404-017-1866-y)

3.       It should be helpful for the readers if the authors could provide the simulation file.

4.       PDMS is difficult to scale up for massive production. Are there alternative materials?

5.       It is difficult to understand why the authors used scheme 1 and then figure 1.

Author Response

Response to reviewer

  1. Schemes 1C and 1D are so small. They need to be enlarged.

Response 1: Thank you for your thoughtful suggestion. We have modified the Schemes 1C and 1D as follows

  1. Reference for equation2(e.g.https://link.springer.com/article/10.1007/s10404-017-1866-y)

Response 2: Thank you for your thoughtful suggestion. We have cited this article (Reference number 29) at equation 2 in the manuscript (to see the line 141 in page 4 and marked in blue).

  1. It should be helpful for the readers if the authors could provide the simulation file.

Response 3: Thank you for your thoughtful suggestion. We will provide simulation files in the appendix for readers to check.

  1. PDMS is difficult to scale up for massive production. Are there alternative materials?

Response 4: For your consideration Hydrogels can be used as alternative materials for PDMS and applied in the fabrication of microfluidic chips.as well.

  1. It is difficult to understand why the authors used scheme 1 and then figure 1

Response 5: We have renamed all the figures in order in the manuscript, as your suggestion.

Reviewer 3 Report

Zhang et al. presented "A dual-channel microfluidic chip for single tobacco protoplast isolation and dynamic capture".  A dual-channel microfluidic chip with multi-capture cavities was constructed in this work. The geometry of the dual-channel microfluidic chip was addressed theoretically. Simulation analysis was used to investigate the capture process of a single cell in a dual-channel microfluidic device. The researchers demonstrated that a single polystyrene microsphere or tobacco protoplast may be isolated and confined in the provided chip. When the inflow flow rate was 0.75L/min, the chip captured 83.33% of the single tobacco protoplast. Furthermore, the dynamic capture of polystyrene microspheres and tobacco protoplasts was shown. Overall, the manuscript is well written. Data presentation is in general OK, but some of the figures needs to be revised to improve their readability. I have the following additional comments:

* The introduction should be improved to better show the limitations of the current literature. 

* The comparison of the device performance to the existing literature (in terms of cost, simplicity, efficiency etc.) should be discussed in more details. 

*Figure 1 can be improved by increasing the text size and maybe font.

*Is clogging an issue for the device operation? It may a good to discuss.

*In the discussion section the following sentence is vague (incomplete) “The theoretical was discussed to design the geometry of this chip”

*What parameters were targeted in the optimization of the geometry of the design?

*If you increase the number of the trapping sites, does the capture efficiency change?

Author Response

Response to reviewer

* The introduction should be improved to better show the limitations of the current literature. 

Response 1: Thank you for your thoughtful suggestion. Based on your suggestions we revised our manuscript (to see the line 51 to 59 and 60 to 61 and 63 to 65 and 70 to 72 in page 2 and marked in blue)

* The comparison of the device performance to the existing literature (in terms of cost, simplicity, efficiency etc.) should be discussed in more details. 

Response 2: Thank you for your thoughtful suggestion. Based on your suggestions we discussed cost, simplicity, efficiency, etc. in the introduction of the manuscript (to see the line 51 to 59 and 60 to 61 and 63 to 65 and 70 to 72 in page 2 and marked in blue).

*Figure 1 can be improved by increasing the text size and maybe font.

Response 3: Thank you for your thoughtful suggestion. We have modified Figure 1 in the manuscript according to your suggestions. This is shown in the figure below.

*Is clogging an issue for the device operation? It may a good to discuss.

Response 4: Thank you for your thoughtful suggestion. The discussion of the causes of clogging is discussed in Section 3.3 of the manuscript (to see the line 218 to 221 in page 7 and marked in blue). Of course local clogging can also be caused by random effects of chip fabrication accuracy, experimental process.

*In the discussion section the following sentence is vague (incomplete) “The theoretical was discussed to design the geometry of this chip”

*What parameters were targeted in the optimization of the geometry of the design?

Response 5: Thank you for your thoughtful suggestion. The geometry of W, Aa, Lh, Lw, Depth of Main channel, and the bypass channel sizes are all our target parameters. These parameters are optimized according to the dimensions of our target cells in our theoretical analysis.

*If you increase the number of the trapping sites, does the capture efficiency change?

Response 6: Thank you for your thoughtful suggestion. From the theoretical analysis, there are many reasons affecting the cell capture efficiency (chip preparation, improper experimental operation, etc.) If the number of trapping points is increased, the single cell capture efficiency will not change much, but the number of captured cells can be increased, which is also an improvement in the single cell capture ability of the microfluidic chip designed by us.

Round 2

Reviewer 1 Report

This paper can be accepted in its present form.

Reviewer 2 Report

I do not have further comments.